# Biocompatibility and Corrosion of Microplasma-Sprayed Titanium and Tantalum Coatings versus Titanium Alloy

Darya Alontseva [1,2,*], Yuliya Safarova (Yantsen) [1,*], Sergii Voinarovych [3], Aleksei Obrosov [4], Ridvan Yamanoglu [5], Fuad Khoshnaw [6], Hasan Ismail Yavuz [5], Assem Nessipbekova [1], Aizhan Syzdykova [1], Bagdat Azamatov [1,2], Alexandr Khozhanov [1,2] and Sabine Weiß [4]

1 Laboratory of Bioengineering and Regenerative Medicine, National Laboratory Astana, Nazarbayev University, Astana 010000, Kazakhstan; assem.nessipbekova@nu.edu.kz (A.N.); syzdykova@nu.edu.kz (A.S.); bazamatov@ektu.kz (B.A.); akhozhanov@ektu.kz (A.K.)
2 School of Digital Technologies and Artificial Intelligence, D. Serikbayev East Kazakhstan Technical University, Ust-Kamenogorsk 070010, Kazakhstan
3 E.O. Paton Electric Welding Institute of NAS of Ukraine, 11 Kazymyr Malevich Street, 03150 Kyiv, Ukraine; serge.voy@gmail.com
4 Department of Physical Metallurgy and Materials Technology, Brandenburg Technical University, 03046 Cottbus, Germany; aleksei.obrosov@b-tu.de (A.O.); sabine.weiss@b-tu.de (S.W.)
5 Department of Metallurgical and Materials Engineering, College of Engineering, Kocaeli University, Kocaeli 41380, Turkey; ryamanoglu@kocaeli.edu.tr (R.Y.); hasanismail.yavuz@kocaeli.edu.tr (H.I.Y.)
6 School of Engineering and Sustainable Development, De Montfort University, Leicester LE1 9BH, UK; fuad.khoshnaw@dmu.ac.uk
* Correspondence: dalontseva@ektu.kz (D.A.); yantsen@nu.edu.kz (Y.S.)

**Abstract:** This study investigates the in vitro biocompatibility, corrosion resistance, and adhesion strength of a gas abrasive-treated Ti6Al4V alloy, alongside microplasma-sprayed titanium and tantalum coatings. Employing a novel approach in selecting microplasma spray parameters, this study successfully engineers coatings with tailored porosity, roughness, and over 20% porosity with pore sizes up to 200 μm, aiming to enhance bone in-growth and implant integration. This study introduces an innovative methodology for quantifying surface roughness using laser electron microscopy and scanning electron microscopy, facilitating detailed morphological analysis of both the substrate and coatings. Extensive evaluations, including tests for in vitro biocompatibility, corrosion resistance, and adhesive strength, revealed that all three materials are biocompatible, with tantalum coatings exhibiting superior cell proliferation and osteogenic differentiation, as well as the highest corrosion resistance. Titanium coatings followed closely, demonstrating favorable osteogenic properties and enhanced roughness, which is crucial for cell behavior and attachment. These coatings also displayed superior tensile adhesive strengths (27.6 ± 0.9 MPa for Ti and 28.0 ± 4.9 MPa for Ta), surpassing the ISO 13179-1 standard and indicating a robust bond with the substrate. Our findings offer significant advancements in biomaterials for medical implants, introducing microplasma spraying as a versatile tool for customizing implant coatings, particularly emphasizing the superior performance of tantalum coatings in terms of biocompatibility, osteogenic potential, and corrosion resistance. This suggests that tantalum coatings are a promising alternative for enhancing the performance of metal implants, especially in applications demanding high biocompatibility and corrosion resistance.

**Keywords:** biocompatible coatings; in vitro test; corrosion resistance; microplasma spraying (MPS); medical implants; coating techniques

## 1. Introduction

In recent decades, the medical implant industry has undergone profound transformations driven by the pursuit of advanced medical implant technologies to reduce implant failure after joint replacement, speed up implant healing, and prolong the life of implants in

the patient's body. Meanwhile, the achievement of implant materials hinges on mechanical behaviors, biocompatibility, and corrosion resistance [1–4].

The biocompatibility of the material implies that after the implant is introduced into a living body, it should maintain the normal cellular activity of the body at the site of implantation without any local and systemic toxic effects on the host tissue; it must be non-immunogenic, osteoconductive (i.e., forming a direct connection with bone tissue, connecting osteogenic cells and providing biological flows), and/or osteoinductive (inducing the differentiation of cells into osteogenic chondrocytes or osteoblasts from surrounding non-bone tissues on its surface, i.e., promoting bone growth) and should also cause the formation of blood vessels in or around the implant [1,5].

Biomaterials function within the body in aggressive environments, such as blood and interstitial fluids containing various proteins, carbohydrates, fatty acids, coenzymes, hormones, salts, and cellular wastes [5]. Clinical studies have indicated localized corrosion of metallic implants under hazardous circumstances [6,7]. It has been found that allergic, poisonous, or carcinogenic ions are released into the body due to corrosion. These metal ions interacting with the tissue induce severe responses, threatening the patient's life and causing revision surgery and implant loosening [8,9]. Therefore, in creating biomaterials, numerous studies must be performed to understand their electrochemical characteristics. In these tests, the material's resistance against corrosion is evaluated by analyzing the characteristics of the oxide film generated on the material surface [5,6].

Currently, titanium alloys are the most common materials used for the manufacture of medical implants due to the alloys' high corrosion resistance and biocompatibility properties, but mainly due to their physical and mechanical properties, in particular the density and elastic modulus (close to values for human bone), making it possible to produce, even via turning or casting methods, relatively lightweight implants capable of withstanding the required loads in the human body [5,6,10]. The failure rate of titanium alloy implants is very low, with more than 89% of titanium alloy implants lasting more than 10 years [2,6,10]. The high corrosion resistance of titanium implants subjected to localized corrosion in the human body is ensured by the formation of oxide films [8], which contribute to the successful osseointegration of titanium implants in areas of compromised bones, as shown by clinical experience [10]. It should be noted that the corrosion resistance properties and biocompatibility of unalloyed titanium are higher than those of its alloys [7]. At the same time, the biological inertness of titanium alloys limits their ability to directly bind to bone tissue, so the biological activity of the surface of titanium alloy implants is typically increased via surface roughening treatments to have a stimulating effect on osteoblast proliferation and differentiation [11]. Thus, the key point is the surface treatment of titanium for implants and porous roughness, which is critical for implant engraftment. Techniques range from sandblasting, which increases surface area for bone integration, to acid etching for micro-pore formation enhance cell adhesion. Anodization further augments corrosion resistance and biocompatibility, incorporating bioactive substances. Laser treatment creates specific patterns for bone integration, while plasma spraying offers precise control over microstructure, including essential porosity and roughness for optimal cell adhesion and implant longevity [3,4].

However, certain concerns of titanium (e.g., hypersensitivity to Ti) have led to the exploration of alternative materials. Tantalum, as a potential metallic biomaterial for implants, has attracted more and more attention due to its excellent anti-corrosion properties and biocompatibility [11–14]. The biocompatibility of tantalum is achieved by forming a corrosion-resistant, relatively thick (about 5 μm) surface oxide layer [7]. However, tantalum's significantly high modulus of elasticity and substantial mechanical incompatibility with bone tissue makes it unsuitable for load-bearing implants manufactured through the aforementioned traditional turning or casting methods [11]. Tantalum has a higher density than titanium (16.65 g/cm$^3$ for Ta and 4.505 g/cm$^3$ for Ti at 20 °C) [15,16] and is more difficult to process. Therefore, at present, either porous implants are made from tantalum via 3D printing, which makes it possible to lower the elastic modulus and reduce the weight

of the implant [17], or tantalum is used as a coating [11,12,14]. The second option may be preferable, given the high cost of tantalum. Thus, at present, tantalum and its alloys are in demand for the production of implants mainly in special cases when patients are allergic or, more precisely, hypersensitive to the traditional metal. However, the situation can be changed for the better by improving technologies for manufacturing tantalum coatings on titanium implants. Thus, while titanium alloys remain the primary material used for orthopedic implants, unalloyed titanium (Ti) and tantalum (Ta) are becoming increasingly promising for implant coatings.

Along with the material, porosity and surface roughness play important roles in influencing the properties of biocompatibility and corrosion resistance of the implant. The surface of the implant first comes into contact with living tissue when the implant is placed in the human body. Therefore, the initial reaction of living tissue to the implant material depends on the properties of the implant surface. To provide the best outcomes, different techniques have been used to manufacture and develop biomaterial applications, such as gas abrasive surface treatment [1,2,4,12], powder technology [18,19], 3D printing technologies [2,4,13,17], vacuum plasma spraying (VPS) [11,12], thermal plasma spraying (TPS) of coatings from biocompatible materials on endoprosthetic implants [20–24], etc.

Nowadays, not only the choice of materials for coating medical implants and endoprostheses but also the optimal porosity and roughness of these coatings are the subject of debate among researchers; a detailed discussion of the problem can be found in the review article by Alontseva et al. [4]. Porosity or surface roughness parameters are not included in international standards for medical implant coatings. Further research is needed to establish the porosity and roughness characteristics that provide the bioactivity and corrosion resistance of specific materials.

It is clear that the surface roughness of the implant influences the responses of the patient's cells and tissues; greater roughness increases the implant's surface area in contact with the bone and, therefore, increases the presence of osteoblasts. This, in turn, improves the fixation of the implant to the bone, which is necessary for bone-fused joint replacements. In particular, for orthopedic titanium implants, the average surface roughness (Ra) recommended by researchers is in a wide range from 0.07 to 100 µm [25,26]. Interestingly, as has been noted by Pligovka et al., 2023, the surface of metals can be modified to create an ordered rough surface 3D two- or three-level column-like systems [27,28]. However, there have been no systematic studies of the effect of surface roughness on biocompatibility.

Regarding porosity and pore sizes, the opinions of researchers also somewhat differ. However, it is generally confirmed that porous coatings with a porosity of at least 20% and with a pore size varying from 50 µm to 300 µm contribute to the reliable fixation of the implant in the bone by increasing the area of contact with bone tissue [29,30]. These porous coatings have a structure similar to that of bone, which allows bone cells and blood vessels to grow into the implant's pores.

These considerations are closely related to the choice of coating technology required to achieve the desired coating characteristics of the selected material and ensure good adhesion to the substrate. This study uses microplasma spraying (MPS) of titanium and tantalum coatings on a titanium alloy. TPS techniques, which include MPS, are widely used in applications related to the metalworking industry [31], but for the biomedical field, it is an innovative topic, the possibilities of which are currently being studied [4,32]. The high temperature (about 20,000 °C) of the plasma jet provides the MPS of metals with high melting points, such as Ti (melting point at normal pressure is 1670 °C) [16] and Ta (melting point at normal pressure is 3017 °C) [15].

An inert gas environment (usually argon, which serves as a plasma-forming and plasma-transferring gas) protects the surface from oxidation during the TPS process. A distinctive feature of MPS is that due to the low power of the process (up to 2 kW), there is no problem with the volumetric heating of the implant during coating spraying. The small size of the plasma spray spot (up to 10 µm) makes it possible to reduce the loss of sprayed material when coating small parts, which include most parts of endoprostheses.

Varying MPS parameters following the matrix of the factorial experiment make it possible to select a combination of spraying parameters that ensures the formation of coatings with specified porosity values [22,23]. The use of a robotic MPS allows for the precise spraying of coatings on implants of complex shapes, moving the plasma source at a constant modulus speed along a 3D trajectory along the surface of the implant with precise adherence to the spraying distance and a plasma jet perpendicular to the surface [23,33]. Previous papers [22,34] have shown that by using robotic microplasma spraying of commercially pure titanium and hydroxyapatite coatings on medical titanium implants, it is possible to obtain coatings with the desired levels of porosity and roughness and satisfactory adhesion to the substrate, including coatings with 95% purity and 90% crystallinity from hydroxyapatite on 3D-printed trabecular Ti6Al4V alloy substrates. It was shown in the article [24] that MPS of tantalum coatings on a titanium alloy is feasible and makes it possible to obtain hard coatings (the average microhardness of microplasma coatings from pure Ta on a Ti6Al4V alloy substrate was $851 \pm 30$ HV, while the microhardness of the substrate was $329 \pm 14$ HV).

Therefore, it follows that studies of the corrosion resistance and biocompatibility of certain materials are necessary to find out about the optimal characteristics of the surface roughness and porosity of implants. At the same time, in vitro testing using animal cells is the most straightforward and most reliable method and often used to measure biocompatibility characteristics, while analysis of polarization curves in simulated saline solution is one of the main methods for evaluating and comparing corrosion resistance performance, which, altogether, make it possible to compare the results obtained in different studies.

In vitro assessment of implant biocompatibility is a critical step in evaluating the effectiveness and safety of medical implants. The choice of cell type depends on the intended application of the implant and the specific biological interactions that need to be assessed. In this regard, mesenchymal stem cells (MSCs) are increasingly recognized as a valuable cell model for in vitro studies for bone-associated research, primarily due to their remarkable ability to differentiate into various cell types, including osteoblasts, which are crucial for bone formation and integration with implants. The use of MSCs offers a more accurate and dynamic model to understand how titanium implants interact with human tissues at the cellular level. This is particularly relevant in the context of osseointegration, where the biological interaction between the implant surface and the surrounding bone tissue is vital for the success of the implant. With MSCs, researchers can closely mimic the in vivo environment, which allows for the detailed investigation of cellular responses to different surface modifications of metal implants. This approach aids in optimizing implant design and enhancing biocompatibility, ultimately improving patient outcomes in orthopedic, dental, and other types of implant surgeries [35].

This study investigates the in vitro biocompatibility and corrosion resistance of gas abrasive-treated titanium alloy Ti6Al4V and microplasma-sprayed unalloyed Ti and Ta coatings on this alloy substrate. This study aims to unveil insights into the selection of the best material, as well as the optimal porosity and surface roughness characteristics, through the scientifically based selection of implant surface treatment parameters to ensure the best biocompatibility and corrosion resistance. Based on these results, it will be possible, firstly, to conclude the preference for the choice of material and method of surface treatment of the implant; secondly, to recommend specific surface treatment parameters that provide the best combination of biocompatibility and corrosion resistance properties; and, thirdly, to contribute to the understanding of the mechanisms influencing coating characteristics, such as porosity and roughness, on the biocompatibility and corrosion resistance of the implant-coating system.



## 2. Materials and Methods

### 2.1. Materials

The materials used for MPS were a wire with a diameter of 0.3 mm made of unalloyed tantalum (Ta) for surgical implant applications; medical-grade EB-2 (electron-beam) produced by "Ulba Metallurgical Plant" ("UMP") JSC, Ust-Kamenogorsk, Kazakhstan, with a composition indicated in Table 1; and a wire with a diameter of 0.3 mm made of commercially pure titanium (CP-Ti) VT1-00, with a composition indicated in Table 2.

**Table 1.** Chemical composition of unalloyed Ta for surgical implant applications, wt. %.

| Standard | Fe | N | O | Nb | C | Ti | W | Mo | H | Ni | Ta |
|---|---|---|---|---|---|---|---|---|---|---|---|
| ASTM F560-17 [36] | ≤0.010 | ≤0.010 | ≤0.015 | ≤0.10 | ≤0.010 | ≤0.010 | ≤0.050 | ≤0.020 | ≤0.0015 | ≤0.010 | Balance |

**Table 2.** Chemical composition of commercially pure titanium VT1-00, wt. %.

| Standard | Al | Fe | Si | C | N | H | O | Ti |
|---|---|---|---|---|---|---|---|---|
| Interstate Standard Group B51 [37] | <0.3 | <0.15 | <0.08 | <0.05 | <0.03 | <0.003 | <0.12 | Balance |

High-purity Ta and CP-Ti wires were sprayed to Ti6Al4V alloy substrates using the standard composition indicated in Table 3.

**Table 3.** Chemical composition of titanium alloyTi6Al4V, wt. %.

| Standard | Fe | N | O | Al | C | V | H | Ti |
|---|---|---|---|---|---|---|---|---|
| ISO 5832-3 [38] | <0.3 | <0.05 | <0.2 | 5.5–6.75 | <0.08 | 3.5–4.5 | <0.015 | Balance |

The specimens used for research were manufactured on computer numerical control (CNC) machines. Specimens in the form of disks with thicknesses of 3 mm were cut from Ti6Al4V titanium alloy rods with diameters of 20 mm on a CTX 510 ecoline CNC machine (DMG MORI AG, Bielefeld, Germany). The shapes and dimensions of the specimens were chosen to be typical for corrosion and biocompatibility tests and to ensure the sufficient thickness of the substrate compared to the coating to avoid overheating the substrate during MPS of the coatings.

### 2.2. Surface Treatment of Ti6Al4V

#### 2.2.1. Gas Abrasive Surface Treatment

Before applying MPS coatings on titanium alloy, the surfaces of the substrates are required to be degreased with acetone and ultrasonically cleaned. To ensure proper coating adhesion, it is important to pre-treat the substrate to increase its roughness. Gas abrasive treatment is also necessary for the chemical activation of the surface. Since the chemical activity of the substrate reduces rapidly due to chemical gas adsorption from the atmosphere and oxidation, the time between gas abrasive surface treatment and coating should not exceed 2 h. The specimens or implants must be stored in a hermetically sealed container (e.g., in an exicator) between the periods of manipulations. The gas abrasive surface treatment was carried out through a Contracor ECO abrasive blasting machine (Comprag Group GmbH, Wuppertal, Germany) using normal-quality A14 electrocorundum $Al_2O_3$ according to the parameters specified in Table 4.

**Table 4.** The parameters of the gas abrasive treatment of Ti6Al4V alloy substrate.

| Parameters | Settings |
|---|---|
| Fraction of abrasive, mm | 0.6 |
| The pressure of compressed air, MPa | 0.6 |
| Distance from the nozzle cut to the treated surface, mm | 100 |
| The incident angle of an abrasive jet on the surface to be treated, degrees | 90 |
| Linear speed of pistol movement, mm/min | 250 |

After gas abrasive treatment to clean the surface from implanted particles of abrasive material, the specimens (Ti6Al4V alloy disks) were subjected to ultrasonic cleaning in medical alcohol for 15 min. It was necessary to clean the surface from embedded abrasive particles since these particles could have a negative impact on the adhesion strength of the coating to the substrate [39].

2.2.2. Microplasma Spraying of Ti and Ta Coatings

MPS of Ti and Ta wire coatings on Ti6Al4V alloy substrate subjected to gas abrasive treatment and cleaning was conducted using the MPS-004 microplasmatron (produced by E.O. Paton Electric Welding Institute, Kyiv, Ukraine) [40]. Argon served as a plasma-forming and -transporting gas for MPS; additional substrate heating was not carried out.

The speed of linear movement of microplasmatron along the substrate was chosen to be 2.3 m per minute [m/min]. This speed was chosen experimentally to ensure plasma spraying of the coating with a uniform thickness. Experiments have shown that this speed of linear movement of the microplasmatron does not lead to disturbances in the plasma jet flow due to air resistance and, therefore, ensures the stability of the MPS process.

The choice of MPS parameters for Ti coatings was based on the analysis of the results of the experiment on MPS of Ti coatings on Ti6Al4V alloy substrate accomplished in a two-level fractional factorial design ($2^{4-1}$) described in detail in the previous paper [22]. The interpretation of the experimental results was carried out using regression analysis methods. The following parameters were selected as variable parameters: electric current (I in amperes [A]), plasma gas flow rate (Q, standard liter per minute [slpm]), spraying distance (H, [mm]), and wire feed rate (Vw, meters per minute [m/min]). The key criterion was the coating porosity, and the dependence of porosity on MPS parameters was established. The relationships between porosity and the spraying parameters have not yet been established for the MPS of a Ta wire. However, the selection of MPS parameters for this study was based on an experiment described in a previous paper [24], in which the porosity and microhardness of Ta coatings sprayed with two different combinations of the above MPS parameters were assessed. The MPS parameters used in this study to achieve a target coating porosity of about 20% for both Ti and Ta coatings are given in Table 5.

**Table 5.** Parameters of microplasma spraying of Ta and Ti wire with Ti6Al4V alloy.

| Coating Material | Current I (A) | Plasma Gas Flow Rate Gp (slpm) | Spraying Distance H (mm) | Wire Feed Rate Vw (m/min) |
|---|---|---|---|---|
| Ta | 31.0 | 4.0 | 30.0 | 3.2 |
| Ti | 16.0 | 2.3 | 40.0 | 3.0 |

*2.3. Coating Microstructure Characterization*

The study of the microstructure and assessment of coating thickness and pore size were performed using a light microscope BX51 (OLYMPUS, Tokyo, Japan) and a scanning electron microscope JSM-6390LV (JEOL, Tokyo, Japan). To assess the porosity of the coatings, the images of their microstructures were processed using the open-source image-processing program ImageJ Version 1.54h (the National Institutes of Health and the Laboratory for Optical and Computational Instrumentation LOCI, University of Wisconsin, Madison, WI, USA) based on the color-difference analysis method shown in Figure 1. The pore sizes and

porosity of the coating were measured (i.e., the percentage of pores in the coating by area per detected pores relative to the entire area of the coating section), identifying inclusions that differed in terms of the shades of gray color or brightness.

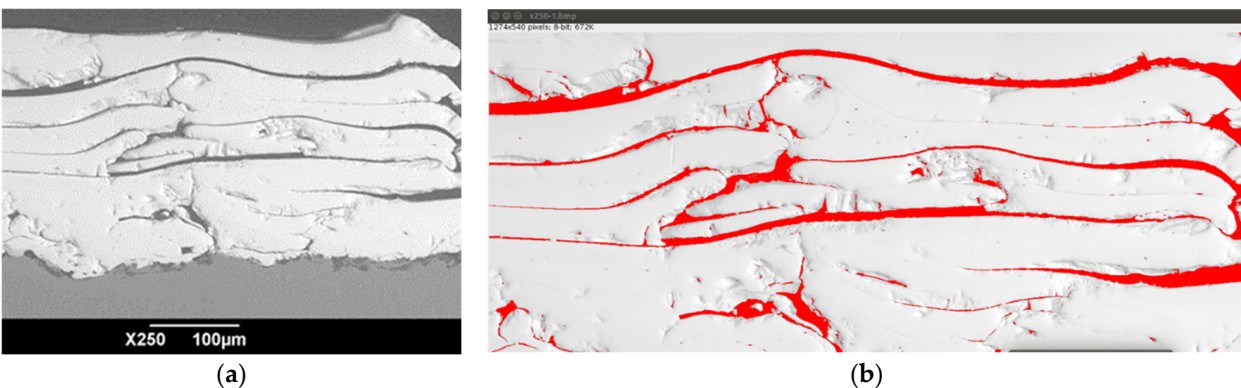

| (a) | (b) |

**Figure 1.** Example image of porosity measurement in a cross-section (**a**) SEM image of a MPS Ta coating; (**b**) same SEM image processed using ImageJ with red fill color.

The measurements were made on the polished cross-section of the coatings following the ASTM E2109-01 standard [41]. To reveal microstructural features, samples of mechanically polished tantalum-coated titanium cross-sections were further subjected to chemical etching for 5 s using a solution of 13 mL of $HNO_3$ + 2 mL of $HF$ + 35 mL of $H_2O$.

### 2.4. Adhesion Strength Test

Adhesion static shear strength testing was performed through a static tensile experiment following the ASTM F1147 standard [42] using a 2054 R-5 (TechMash, Neftekamsk, Russia) universal mechanical testing complex with equipment for testing the adhesion strength of the coatings to a substrate. Following ASTM F1147, a complete, assembled test assembly consisted of two solid parts, one with a coated surface and one with an uncoated surface. The substrate area upon which the coating was applied was calculated as $4.91 \pm 0.01$ cm$^2$. The thickness of the plasma-sprayed coatings was $250 \pm 30$ μm. Coated specimens were glued with counter-specimens using a two-component viscous-flow adhesive of the VK-9 brand (Khimprom, Pervomaisk, Ukraine) with the immovable fixation of the specimens with a compression force of 0.01 MPa, keeping them in this state at a temperature of 60 °C for at least 1 h until the glue polymerized completely. Further, the test was carried out at room temperature; 5 glued samples for each material were tested, and the results were subjected to standard statistical processing.

### 2.5. Surface Roughness Assessment

The surface roughness was analyzed using a confocal laser scanning microscope Keyence VK-X1000 (Neu-Isenburg, Germany). The surface roughness parameters Sa and Sq were measured according to ISO 25178 [43]. Sa is the difference in the height of each point compared to the arithmetical mean of the surface, and the related Sq parameter is the root mean square value of the ordinate values within the definition area. Sa is generally used to evaluate surface roughness, while Sq is equivalent to the standard deviation of heights. The area roughness parameters were chosen since they give more significant values than line parameters. The scan area for each measurement was approximately $700 \times 500$ μm$^2$. Three different regions were randomly selected for the surface roughness measurement of each sample to ensure the reproducibility of the measurements.

### 2.6. Corrosion Test

The polarization test was performed in a three-electrode cell, which was controlled using the AMETEK brand VersaSTAT 4 model potentiostat/galvanostat. In this configuration, the samples acted as the working electrode, while platinum functioned as the reference electrode. Before the polarization test, the samples were immersed in simulated body fluid (SBF), i.e., into a solution with an ion concentration close to that of human blood plasma, for 1 h to attain equilibrium.

The Tafel extrapolation technique was applied to determine the corrosion current density ($i_{corr}$) and corrosion potential ($E_{corr}$). Polarization curves were obtained in a 4 M NaCl solution at 20°. The test comprised a voltage range from $-250$ mV to $+250$ mV vs. a Saturated Calomel Electrode (SCE) with a scan rate of 1 mV/s. Additionally, the area of the samples in contact with the 0.9% NaCl solution was set to 1 cm$^2$.

### 2.7. Cell Viability

Alamar Blue protocol (DAL1025, Thermo Fisher, Waltham, MA, USA) was used to assess the cytotoxicity. In brief, $6 \times 10^3$ MSCs were seeded into each well and incubated for 48 h with implant-enriched media (3-day extract) after adding 10 µL of Alamar Blue reagent was added to each well and incubated for 2 h. Subsequently, absorbance was measured at 570 nm and 600 nm for normalization with a microplate reader Synergy H1 (Bio Tek, Winooski, VT, USA).

### 2.8. Cell Proliferation

CCK8 (96992, Sigma Aldrich, St. Louis, MO, USA) was used to evaluate the proliferation of rat MSCs in vitro. In brief, $5 \times 10^3$ MSCs were seeded into each well of a 96-well plate and incubated for 72 h with implant-enriched media (3-day extract). Once incubated, 10 µL of CCK-8 solution was added to each well and incubated for 4 h. After absorbance, it was measured at 450 nm using a microplate reader Synergy H1 (Bio Tek, Winooski, VT, USA). A calibration curve was used to determine the number of cells. A control group was incubated in the plain DMEM media with no implant extracts.

### 2.9. Alkaline Phosphatase Activity

MSCs were seeded at $8 \times 10^4$ cells per well density to the 48-well plate and cultured until 90% confluency was reached. Further, cells were treated with implant-enriched osteogenic media (3-day extract) for 2 weeks. Then, cell medium supernatants were collected, and ALP enzyme levels were measured using an Alkaline Phosphatase Assay Kit (ab83369, Abcam, Boston, MA, USA). The ALP enzyme of known concentration was plated in a serial dilution to create a standard curve. pNPP substrates were added to each standard and sample well and incubated for 60 min at room temperature. ALP enzyme converts pNPP substrates to a colored p-Nitrophenol (pNP). After incubation, a stop solution was added, and the absorbance was recorded at OD 405 nm with a microplate reader Synergy H1 (Bio Tek, Winooski, VT, USA). The group with plain DMEM media served as a negative control to confirm osteogenesis. No treatment group served as a control to assess the effects of coated implants.

### 2.10. Alizarin Red S Staining

MSCs were seeded at a density of $8 \times 10^4$ cells per well to the 48-well plate and cultured until 90% confluency was reached. Then, cells were treated with implant-enriched osteogenic media (3-day extract) for 5 weeks. At week 5 of differentiation, cells were washed and then fixed with 500 µL 4% formaldehyde for 30 min and stained with 500 uL 2% Alizarin Red S (A5533, Sigma Aldrich, St. Louis, MO, USA) for 1 h, before being washed with MulliQ water 5 times for 30 min. Images were taken using the inverted microscope at 4× magnification (Zeiss Axio Observer, Jena, Germany). After washing, the stain was extracted from the differentiated monolayer with 400 µL 10% acetic acid and heated to 85 °C for 10 min. Further, the slurry was centrifuged to 20,000 rcf for 15 min. The su-



pernatant was collected, and the pH was adjusted to 4.1–4.5 using 100 μL of ammonium hydroxide. Then, the supernatant was transferred to a microplate reader (Bio Tek Synergy H1, BioTek Instruments, Inc., Winooski, VT, USA) and measured at OD 405 nm.

### 2.11. Scanning Electron Microscopy (SEM) Sample Preparation

Coated titanium disks were seeded with MSCs, and cultures in complete DMEM for 48 h. The protocol used for staining was adopted and modified from Geekiyanage et al. [44] Then, cells were fixed with 2% paraformaldehyde for 10 min and stained with 1 M Osmium Tetroxide for 1 h. After, cells were rinsed twice with Milli-Q water for 10 min, dehydrated with Ethanol (from 70% to 100%), and incubated with HDMS (hexamethyldisilazane for 30 min. Following 30 min of drying, samples were covered with 20 nm layer of gold.

An assessment of the sample's topography was performed using a MIRA II scanning electron microscope from Tescan (Brno, Czech Republic). Furthermore, it was used to characterize samples after in vitro tests. Surfaces before observation were coated for 60 s with gold using the sputter coater 108 auto (voltage 40V, current 30 mA) from Cressington Scientific Instruments (Walford, UK).

### 2.12. Statistical Analysis

Data analysis was conducted using SigmaPlot v14 (Systat Software, Inc., San Jose, CA, USA), with all results presented as the mean $\pm$ standard deviation from three separate experiments. To assess differences between group means, a two-way ANOVA statistical test, followed by a Bonferroni post-test, was utilized, considering differences significant at $p < 0.05$.

## 3. Results and Discussion

### 3.1. Coating Microstructure Characterization

Light microscopy images of cross-sections of titanium and tantalum microplasma-sprayed coatings on the Ti6Al4V alloy substrate are shown in Figure 2.

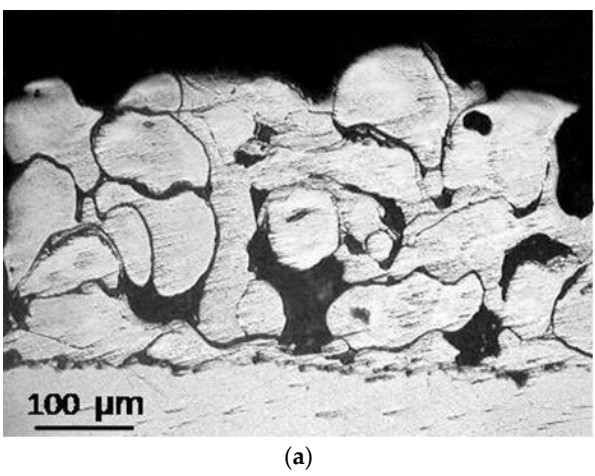 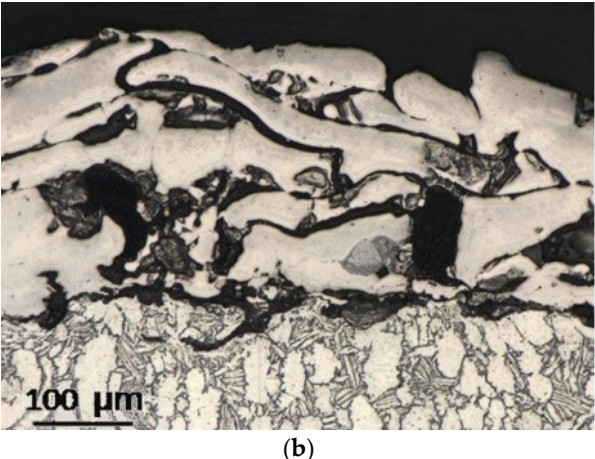

(**a**)                                                                                 (**b**)

**Figure 2.** Microstructure of the Ti coating (**a**) and Ta coating (**b**) on a Ti6Al4V alloy substrate (cross-sections).

The results of measuring the porosity of coatings, indicating the pore sizes, the roughness of the substrate and coatings and the adhesion strength of coatings to the substrate are summarized in Table 6.

**Table 6.** Results of measurements of porosity, roughness, and adhesion strength of the microplasma-sprayed coatings and Ti6Al4V substrate.

| Material/Characteristics | Ta Coating | Ti Coating | Ti6Al4V Alloy after Gas Abrasive Treatment |
|---|---|---|---|
| Porosity | up to 20 ± 2% pore sizes from 20 μm to 200 μm | up to 22 ± 2.5% pore sizes from 20 μm to 200 μm From this point of view, it is feasible to | N/A |
| Mean surface roughness (Sa), μm | 16.4 ± 0.5 | 27.6 ± 2.6 | 4.6 ± 0.1 |
| Root mean square roughness (Sq) | 21.0 ± 1.1 | 34.4 ± 3.5 | 5.8 ± 0.1 |
| Mean static tensile strength of the coating, MPa | 28.0 ± 4.9 | 27.6 ± 0.9 | N/A |

As follows from the results of assessing the microstructural characteristics of coatings given in Table 6, when using the MPS parameters indicated in Table 5, similar values of coating porosity were achieved for both coatings (porosity of at least 20%, with sufficiently large pores up to 200 μm), which is desirable for coatings of endoprosthetic implants. At the same time, the adhesive strengths of the coatings turned out to be quite satisfactory.

*3.2. Adhesion Strength*

The adhesion strength of the Ti coating was 24.7 ± 5.7 MPa (Table 6), which met the requirements of ISO 13179-1 for plasma-sprayed unalloyed titanium coatings on metallic surgical implants [45]. According to this standard, the mean static tensile strength of the surface coating was greater than 22 MPa.

Although there appears to be no international standard for plasma-sprayed Ta coatings on metallic surgical implants (the authors found no such standard), this study found that the adhesion strengths of the Ta microplasma-sprayed coatings to a Ti6Al4V alloy substrate were also more than 22 MPa, namely the mean static tensile strength of the Ta coating was 28.0 ± 4.9 MPa (Table 6), which is even better than that of the Ti coatings. Thus, the analysis of the adhesive strengths of the microplasma-sprayed Ta and Ti coatings on a Ti6Al4V alloy indicates the possibility of using MPS technology for coatings of metallic surgical implants. Specimens of Ti and Ta coatings after the tensile adhesion test are shown in Figure 3.

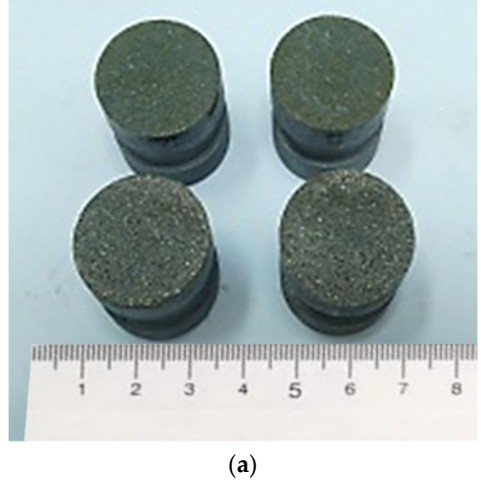

(**a**)

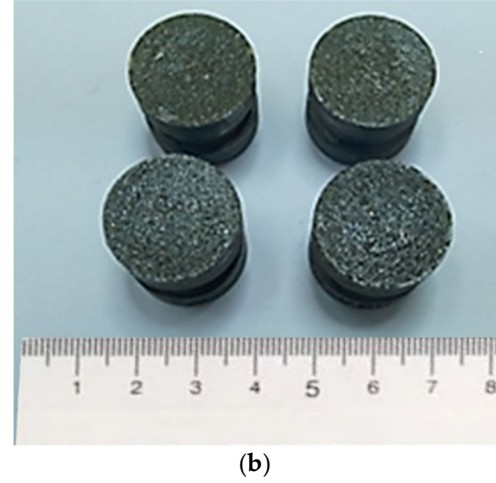

(**b**)

**Figure 3.** Specimens of Ti (**a**) and Ta (**b**) coatings after the tensile adhesion test are shown.

Notably, in 80% of cases, the damage exhibited adhesive characteristics. As is illustrated in Figure 2, the coatings remained largely intact during the tensile adhesion test. The detachment occurred along the glue itself, which indicates the good adhesion of the coating to the substrate and strong adhesion of the coating particles to each other. Such a relatively high adhesion strength of the coating was most likely due to the correct choice of the MPS

parameters and the use of grit blasting, which can be used to achieve maximum coating adhesion [46].

### 3.3. Surface Roughness

3D laser scanning images of the surface of all three materials are presented for comparison in Figure 4. In this work, Sa and Sq roughness were measured. Sq provides a more comprehensive surface roughness analysis compared to Sa (arithmetic mean roughness). Sq considers the distribution of heights across the entire surface, considering both the high and low points. Sq is more sensitive to extreme surface height variations compared to Sa. High peaks and deep valleys contribute more significantly to Sq, making it a valuable parameter for identifying localized roughness issues or irregularities in the surface that might not be captured as effectively by Sa.

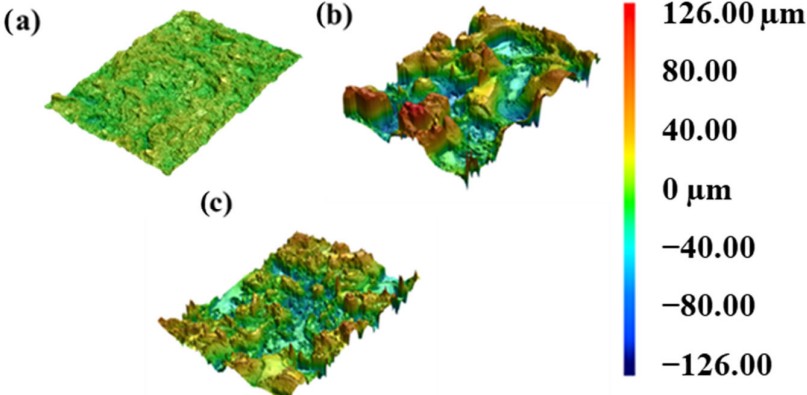

**Figure 4.** 3D laser scanning of the surface: (**a**) Ti6Al4V alloy and MPS coating; (**b**) Ti; (**c**) Ta ($700 \times 500$ μm$^2$).

In contrast, Sa only looks at the average height deviations, potentially missing out crucial information about the surface texture. The substrate sample revealed the lowest roughnesses of $4.6 \pm 0.1$ μm (Sa) and $5.8 \pm 0.1$ μm (Sq), respectively (Figure 4a). In contrast, sprayed coatings revealed a significant increase in roughness of up to 5 times (Figure 4b,c). Sa and Sq revealed a similar trend in roughness. Ti spraying led to an increase in roughness, and the highest value for titanium coating was achieved ($27.6 \pm 2.6$ μm (Sa) and $34.4 \pm 3.5$ μm (Sq)). At the same time, Ta coating revealed a 40% lower roughness ($16.4 \pm 0.5$ μm (Sa) and $21.0 \pm 1.1$ μm (Sq), respectively). Grit-blasted specimens in this work have comparable roughness values to those in the results from Lewallen et al., (Ra $5.74 \pm 0.19$ μm) and higher than those of bead-blasted substrates (Ra $1.10 \pm 0.18$ μm) [26]. The surface roughness strongly affects cell morphology, proliferation, and surface energy. Rough topography promotes tissue growth and biocompatibility. Biocompatibility is limited by flat implant surfaces, which hinder suitable cell adhesion. On the other hand, Borsari et al. [47] investigated the influence of roughness on cell proliferation. They confirmed that the surface morphology strongly affects cell behavior and too high roughness (over 40 μm) leads to decreased cell proliferation.

Figure 5 represents the surface morphology of substrate-, titanium-, and tantalum-sprayed coatings. The substrate surface after gas abrasive treatment exhibits a morphology usual for this kind of treatment. Al$_2$O$_3$ particles could be recognized on the surface, some of which are embedded into the substrate. Al element has a controversial influence on health, but some groups report that biocompatibility could decrease with an increase in Al content [48], and aluminum could cause Alzheimer's disease [49].

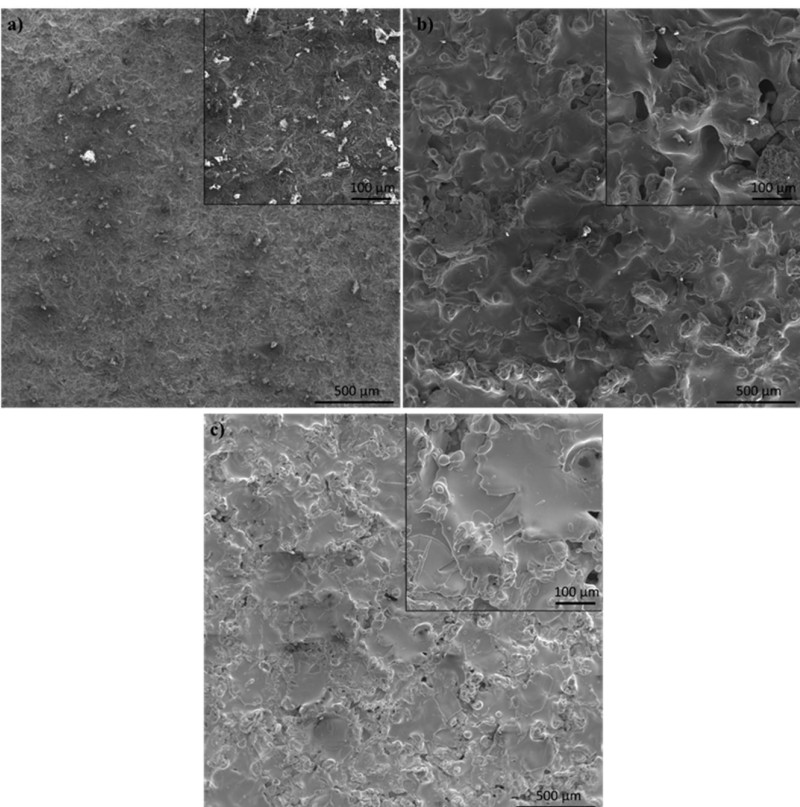

**Figure 5.** SEM images of the surface of the (**a**) substrate, (**b**) Ti, and (**c**) Ta coatings.

Both coatings have a rough and irregular surface with uniformly distributed pores and partially/completely melted particles. Surface roughness is usually divided into three levels depending on the scale of the features, which include nano- (1–100 nm), micro- (1–10 μm), and macro-sized (over 10 μm) topologies [50]. Compared to smooth surfaces, both coatings reflect a macroscale degree of high roughness, which enhances the mechanical stability of prostheses in the long term. Meanwhile, because of the increased bacterial adherence, this may result in peri-implantitis [51]. On the surface, pores of different sizes could be observed. Mostly, they are under 50 μm on the surface, which could lead to fibrovascular in-growth. The surface of Ta coating was practically crack-free, whereas in the case of Ti coating, some microcracks could be observed, which might affect the mechanical properties and adhesion of coatings.

As previously demonstrated by the authors [4], coatings can be categorized into three distinct groups based on the MPS parameters, particularly the particle heating level and impact velocity upon the substrate:

- Group 1 involves scenarios where particles are fully melted before reaching the substrate. This results in the formation of dense coatings with an average porosity below 4%.
- Group 2 describes situations where particles are partially solidified as they approach the substrate, along with fully molten particles. The resulting coating structure is typically porous, with an average porosity of around 8%.
- Group 3 encompasses coatings created from particles that have begun to solidify and are moving at a slower speed upon reaching the substrate. These conditions lead to the formation of coatings with the highest average porosity, approximately 20%, and feature large pores ranging from 20 μm to 200 μm in size. Coatings with such pore sizes in endoprostheses can facilitate the in-growth of blood vessels into the coating's pores. This process is beneficial for bone tissue formation and nourishment, thereby improving the fixation and osseointegration of the endoprosthesis within the human body.

Thus, it becomes clear how the selected parameters of MPS contributed to achieving the desired porosity characteristics. They facilitated the formation of coatings belonging to Group 3.

### 3.4. Corrosion Analysis

A polarization test was applied to the Ti6Al4V alloy treated with gas abrasive and unalloyed Ti and Ta coatings coated on this alloy using the MPS technique, and the corrosion resistance values were measured. The polarization curve graph is given in Figure 6. The corrosion potential in the diagram is the potential of the corroding surface in an electrolyte with respect to a reference electrode. This value is determined via Tafel's extrapolation of the anodic and cathodic curves in the potentiodynamic polarization diagram. The corrosion current density is similarly measured using potentiodynamic polarization curves and highly connected to the corrosion rate. The better corrosion performance of the material is attained by raising $E_{corr}$ and reducing $i_{corr}$ [52].

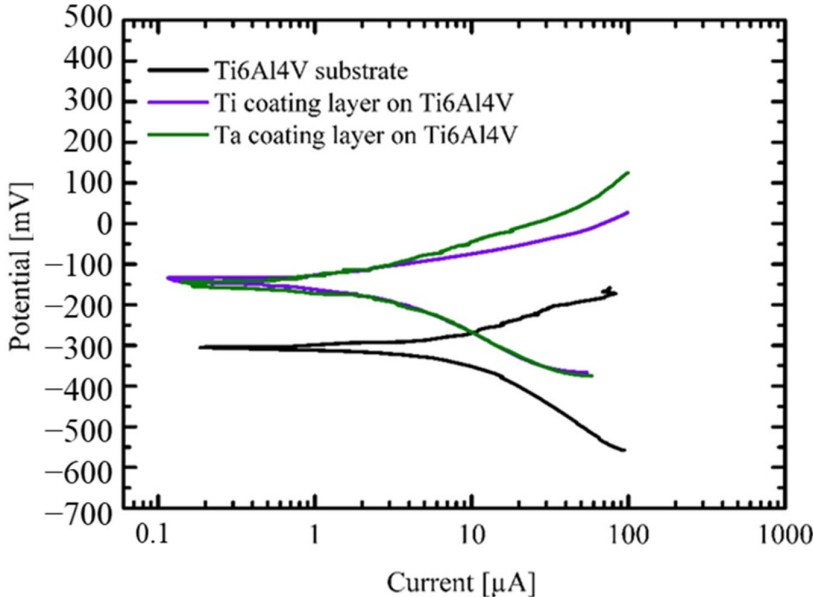

**Figure 6.** Tafel extrapolation graph of the samples.

As observed, the maximum corrosion current density and the lowest corrosion potential values were obtained from the Ti6Al4V sample. On the other hand, the polarization curve of the Ti-coated sample shows that the $i_{corr}$ value decreased and the corrosion potential increased compared to Ti6Al4V. This indicates that the corrosion resistance of the Ti-coated sample was enhanced compared to Ti6Al4V. On the other hand, the lowest $i_{corr}$ and the highest $E_{corr}$ values were measured in the Ta-coated sample. Therefore, the Ta-coated sample showed the best performance in terms of corrosion resistance in the study. One of the main reasons for the excellent corrosion properties of Ta is the naturally occurring $Ta_2O_5$ (5 μm thick) surface oxide layer [24,53,54]. This extremely stable layer provides excellent corrosion–erosion resistance in highly acidic conditions without substantial weight or roughness changes compared to titanium and stainless steel implants [55]. As has been noted by Pligovka et al., 2023, the surface of implants should be protected by oxide, preferably nanostructured oxide [27].

In the literature, many studies have demonstrated the superior corrosion resistance properties of Ta and Ta-based alloys in various acidic and basic solutions. For instance, Zitter and Plenk [56] evaluated the current densities of several implant metals and alloys to assess the rate of electrochemical interactions between implant and tissue. A decrease in current density suggests an improvement in corrosion resistance. The current densities decreased in the following order: steels < CoCr alloy < Ti-6Al-4V < Ti < Nb < Ta. As

can be seen, the lowest current density was measured in Ta. This is associated with a stable surface oxide layer at the implant–tissue interface, which prevents electron exchange and subsequent redox (oxidation-reduction) reactions. Similarly, Hee et al. [57] coated Ta on Ti6Al4V substrate via a sputtering method and compared the corrosion resistance of the coating with that of the substrate. The corrosion current density of the Ta-coated Ti6Al4V decreased from $2.0 \times 10^{-8}$ to $3.4 \times 10^{-9}$ A·cm$^{-2}$ compared to the uncoated Ti6Al4V substrate. Further research confirms that the microplasma-sprayed unalloyed Ta coating improves the material's corrosion resistance compared to the microplasma-sprayed unalloyed Ti coating and Ti6Al4V [53,58,59]. From this point of view, it is feasible to claim that the results produced in this study are compatible with those of the literature [60]. On the other hand, a key aspect of our research was to ascertain that even the porous coatings after gas abrasive treatment developed through MPS effectively safeguard against corrosion. This is crucial for their application in medical implants and endoprostheses. Our findings suggest that tantalum, in comparison to titanium, emerges as the more favorable material for such coatings. The most intriguing aspect moving forward is to conduct a comprehensive comparison of their biocompatibility.

### 3.5. In Vitro Tests

The reaction of the host to inserted materials and devices is affected by both the material's design and the host's local and systemic conditions. The initial response typically mirrors the established sequence of wound healing stages, ranging from hemostasis to the formation of scar tissue. The impact of the implanted material on this process can be either beneficial or detrimental, leading to outcomes like constructive tissue remodeling, ongoing inflammation, foreign body reaction with encapsulation, or an adaptive immune response [61]. Cell viability involves evaluating how living cells interact with and survive in the presence of implant materials. The goal is to ensure that the implant does not adversely affect the cells and supports or enhances the natural healing process.

The Alamar Blue assay, employed to assess the cytotoxicity, utilizes a responsive dye that shifts in color corresponding to variations in cellular metabolic activity. In our study, this assay revealed a notable enhancement in cell viability—a 65% increase when compared with the Ti6Al4V alloy, as illustrated in Figure 7. This finding indicates that the tested material exhibits a promising level of biocompatibility. However, it is crucial to note that the observed difference in cell viability between our sample and the standard Ti group did not reach statistical significance. This suggests that while the Ti coating shows improved performance in terms of supporting cell growth, its efficacy in this respect is comparable to that of the Ti6Al4V alloy under experimental conditions.

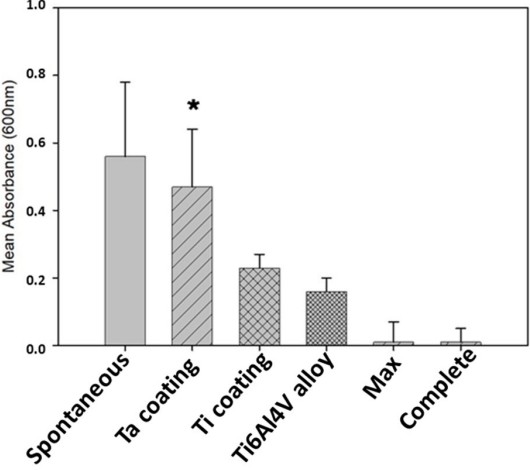

**Figure 7.** Cell viability assay with Alamar Blue. Mesenchymal stem cells were cultured in implant-enriched media (3-day extract) for 48 h. The Alamar Blue protocol (DAL1025, Thermo Fisher) was used to evaluate the number of proliferated cells. *—$p$ value $\leq 0.05$.

To evaluate cell proliferation in the presence of the implant, we employed the CCK-8 assay from Sigma-Aldrich (St. Louis, MO, USA). This assay revealed that both Ta and Ti coatings facilitated a significant enhancement in the cell proliferation rates only on day 3, with an approximate 32% increase compared to the control group, which had no implant extracts (see Figure 8). Furthermore, the Ti6Al4V alloy demonstrated a noteworthy 23% increase in proliferation, statistically significant when compared to the control group. However, when directly comparing the coatings group with the Ti6Al4V alloy, it was observed that only the Ta coating exhibited a statistically significant advantage, showing a 12% higher proliferation rate. These data suggest that Ta might offer superior biocompatibility and cell proliferation properties compared to standard Ti alloys under the tested conditions. The absence of a notable effect on cell proliferation during the initial two days of our study aligns with the findings by Tang et al., 2013 [62]. In their research, the Ta coating's beneficial impact on cell proliferation also only became evident from the third day onward. This temporal pattern suggests that the Ta coating may exert its positive effects on cell growth in a delayed manner rather than immediately upon exposure. This delayed response is a crucial aspect to consider in understanding the dynamics of cell interaction with tantalum-coated materials, especially in the context of implant biocompatibility and effectiveness in promoting tissue integration [63]. The high viability of the coatings could also be related to a higher surface roughness and reference area ratio, which characterize the level of surface expansion. However, the higher Ta proliferation may be attributed to the roughness of titanium being too high, and optimal values have been achieved for tantalum.

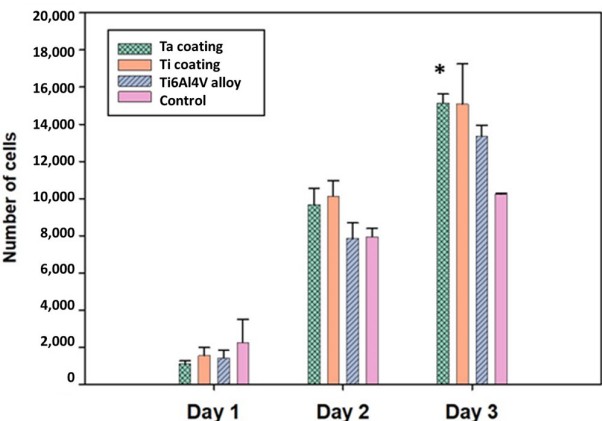

**Figure 8.** Proliferation Assay with CCK-8 during 3-day period. Mesenchymal stem cells were cultured in implant-enriched media (3-day extract) for 72 h. CCK-8 (96992, Sigma Aldrich, St. Louis, MO, USA) was used to evaluate the number of proliferated cells. *—$p$ value $\leq 0.05$.

Assessing cell survival and behavior over extended periods in contact with an implant is vital for a comprehensive understanding of its long-term biocompatibility, potential impact on tissue integration, and healing processes [64]. In this context, evaluating osteogenic capacity is particularly useful, as it provides crucial data on the implant's ability to facilitate successful osteointegration. Our studies of early osteogenic activity with alkaline phosphatase ELISA, conducted after two weeks of differentiation, showed no statistically significant difference between the Ta and Ti coatings and Ti6Al4V alloy. However, both the Ta and Ti coatings demonstrated a positive effect on cell growth compared to the control group, which was not exposed to any implant surface. Specifically, these coatings enhanced the cell proliferation rate by approximately 10% (Figure 9). These results are consistent with earlier data indicating that titanium implants also enhance the osteogenic potential of the cells [65].

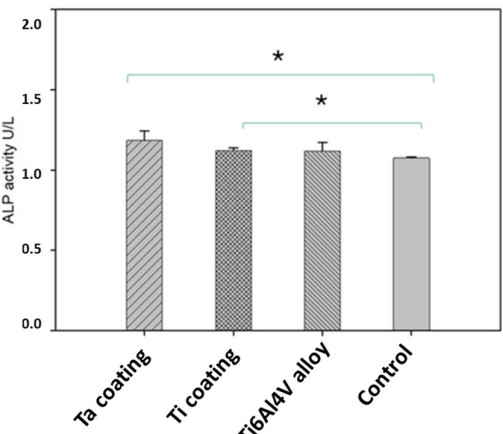

**Figure 9.** Alkaline Phosphatase Assay. Mesenchymal stem cells were cultured in implant-enriched osteogenic media (3-day extract) for two weeks. The Alkaline Phosphatase Assay Kit (ab83369, Abcam, Boston, MA, USA) was used to assess the osteogenic differentiation. *—$p$ value $\leq 0.05$ compared to the control group.

This finding underscores the importance of considering not only the immediate responses but also the prolonged biological responses to implant materials, especially in the context of their application in bone tissue engineering.

For the late-stage assessment of osteogenic differentiation, we used Alizarin Red S staining, a well-established method for evaluating calcium deposition, a hallmark of osteogenesis. In our study, each of the tested groups—the Ta coating, Ti coating, and Ti6Al4V alloy—demonstrated a statistically significant enhancement in osteogenic differentiation compared to the control group, which received no treatment. The increases were 37%, 26%, and 29% for the Ta, Ti, and Ti6Al4V alloy groups, respectively (Figure 10). Notably, when comparing coatings to the Ti6Al4V alloy, only the tantalum coating demonstrated a statistically significant increase in osteogenic differentiation, i.e., 11%. This finding highlights the ability of the tantalum coating to promote osteogenesis compared to standard titanium alloys. It could have significant implications for the development of advanced biomaterials in bone tissue engineering and implant design.

The way that cells adhere to bone grafts continues to be a subject of significant interest. SEM has been consistently recognized as the most effective method for imaging cells on material surfaces. Previous studies [66–68] demonstrated the interaction of bone-like cells with titanium surfaces during the adhesion process. Cells can detect the three-dimensional contours of the substrate that they grow on and adapt to its topography thanks to a flexible cytoskeleton. In this regard, the fluctuation in surface roughness provides the difference in the strength of cell adhesion and, subsequently, its main cell functions, such as proliferation, migration, and differentiation [69]. Cell attachment on the disks after 2 days of culture was evaluated via SEM. In Figure 11, mesenchymal stem cells are shown to adhere inside the pores (shown with arrows). The cells revealed a spindle fibroblast-like morphology typical of MSCs.

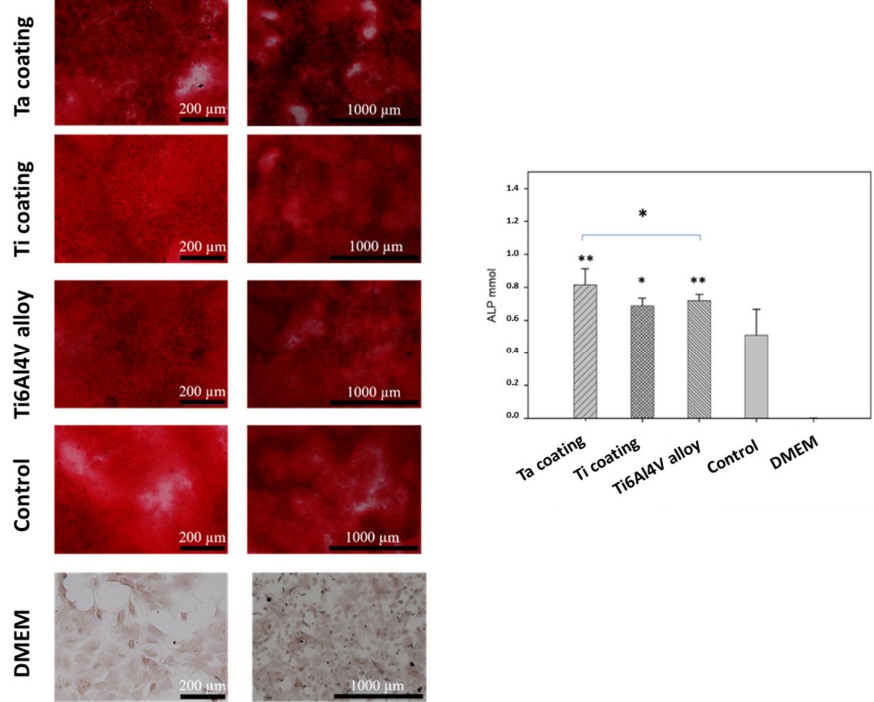

**Figure 10.** Alizarin Red S Assay. Mesenchymal stem cells were cultured in implant-enriched media (3-day extract) for 72 h. CCK-8 (96992, Sigma Aldrich, St. Louis, MO, USA) was used to evaluate the number of proliferated cells. *—$p$ value ≤ 0.05; **—$p$ value ≤ 0.005.

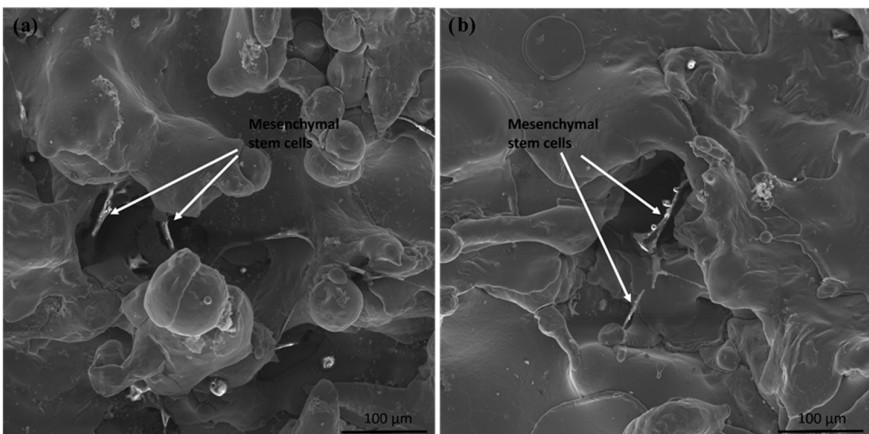

**Figure 11.** SEM images showing MSCs (**a**) inside the pore and (**b**) between the pore's walls of the titanium-coated implant surface.

Figure 12 represents the SEM images showing the attachment, growth, and spreading of cells after culturing for 48 h.

In comparison to the uncoated surface where only some parts of the surface were observed cells, significantly higher attachment and spreading were observed. Filopodia and/or lamelliopodia appear on the surface of the coated samples, indicating that the cells have begun to grow and spread. There is no cytotoxicity effect observed on all surfaces. A similar observation was made by Kuo et al. [51] for graded porous tantalum coatings deposited via vacuum plasma spraying. It could be concluded that the surfaces of the sprayed coatings are more beneficial for the enhancement of the biocompatibility of the Ti-based alloys.

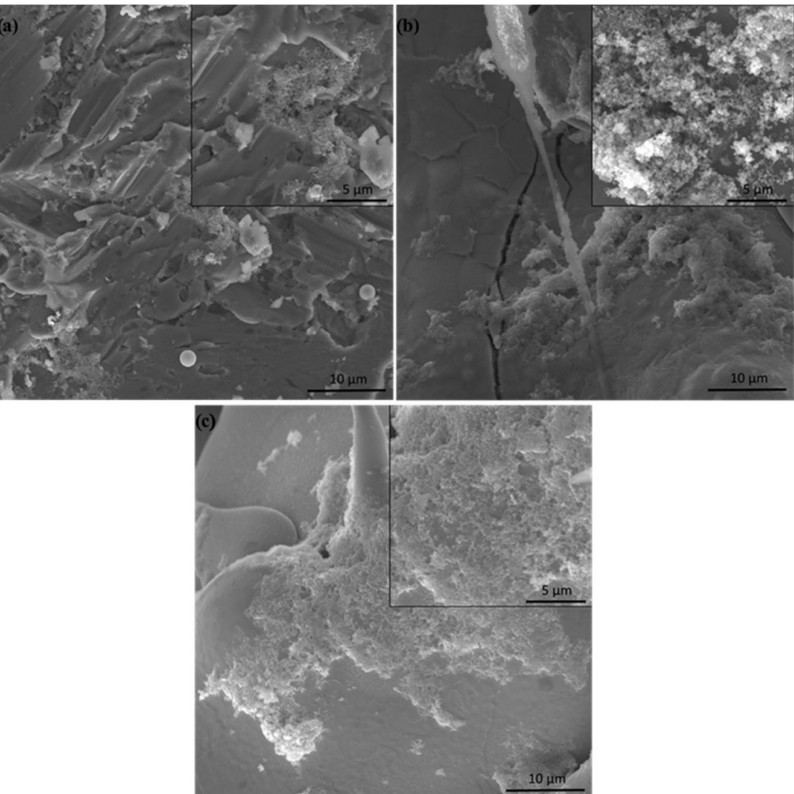

**Figure 12.** SEM images of (**a**) uncoated and (**b**) Ti- and (**c**) Ta-sprayed coatings after 48 h of cell growth.

## 4. Further Perspectives

Therefore, all coatings exhibited satisfactory adhesion to the substrate, were biocompatible, and had good corrosion resistance. However, the Ta coating appeared to have the most significant potential for enhancing the biocompatibility and corrosion resistance of all the titanium implants. In the future, it would be promising to investigate further the effects of MPS parameters on the adhesion strength and surface roughness of coatings, as well as the influence of surface roughness on coating biocompatibility and corrosion resistance.

Future research is planned on the selection of the optimal parameters for MPS of tantalum wire on a titanium alloy using the methods of planning a factorial experiment and the regression analysis of experimental results. Optimal parameters mean those that will allow effective spraying (with minimal loss of material) of a coating with the specified characteristics of porosity and, possibly, surface roughness.

Further studies of biocompatibility will be conducted, with a particular emphasis on the SEM analysis of focal adhesion. This analysis is crucial for assessing the mechanical interactions between cells and their surrounding environment. Understanding how cells attach, spread, and communicate in this context is vital for evaluating the performance of biomaterials in regenerative medicine. Additionally, in vivo studies utilizing the developed models for bone defects will be integral to this research. These studies will provide critical insights into the effectiveness, safety, and clinical applicability of the materials and technologies being developed. Furthermore, these studies will deepen our understanding of the relationship between the architecture and compositions of coatings and cell function in the healing process. This includes how different surface textures, porosities, and material compositions influence cell behavior, such as adhesion, proliferation, and differentiation—all of which are key factors involved in the healing of bone and other tissues.

The combination of SEM analysis of focal adhesions and in vivo studies will provide a more comprehensive understanding of how these innovative materials interact with biological systems. It will help to design and develop more effective and safe biomaterials

for medical applications, as the ultimate goal is to improve patient outcomes by enhancing the body's natural healing processes through advanced biomedical technologies.

## 5. Conclusions

In conclusion, this study provides interesting insights into the in vitro biocompatibility and corrosion resistance of gas abrasive-treated Ti6Al4V alloys and microplasma-sprayed titanium and tantalum coatings. This study selected microplasma spray parameters to achieve coatings with optimal porosity and roughness. Both Ta and Ti coatings achieved a porosity of at least 20% and a pore size of up to 200 μm. This porosity and pore size are considered optimal for such coatings, as they potentially enhance bone ingrowth and osseointegration, critical for the stability and longevity of implants.

Furthermore, this study reported notable tensile adhesive strengths for both types of coatings. The titanium coatings exhibited a strength of $27.6 \pm 0.9$ MPa, while the tantalum coatings showed a slightly higher strength of $28.0 \pm 4.9$ MPa. These values are indicative of a robust bond between the coatings and the alloy substrate, surpassing the minimum requirement set by the ISO 13179-1 standard, which stipulates a tensile adhesion strength of at least 22 MPa for titanium coatings on metal surgical implants.

This study revealed that surface roughness is a critical factor influencing cell behavior and proliferation in the context of biomaterials. Two specific parameters, Sa and Sq, were identified as effective metrics for characterizing the surface roughness of these materials. This distinction is important as it allows for a more precise and standardized assessment of surface textures in biomaterials research. Interestingly, both the titanium and tantalum coatings exhibited significantly higher surface roughness compared to the Ti6Al4V alloy surface after gas abrasive treatment, with an up to five time increase. Among these, the titanium coatings showed the highest roughness values, with Sa recorded at $27.6 \pm 2.6$ μm and Sq at $34.4 \pm 3.5$ μm. Enhanced roughness in titanium coatings is particularly noteworthy, as it implies a greater potential for improving cell attachment and proliferation, which are key factors in the integration and success of medical implants.

In the corrosion analysis conducted, it was observed that the Ti6Al4V alloy subjected to gas abrasive treatment exhibited the lowest corrosion resistance among the tested materials. In contrast, both the titanium- and tantalum-coated samples demonstrated enhanced corrosion resistance compared to the untreated Ti6Al4V alloy. Furthermore, the most notable finding was that the tantalum-coated specimen registered the lowest current density value. This is a key indicator of corrosion resistance, with lower current densities reflecting a more corrosion-resistant material. This result highlights the effectiveness of the tantalum coating in significantly improving the corrosion performance of the base material.

In vitro studies showed a stable positive effect of tantalum coating in proliferation and osteogenic differentiation. Tantalum coating significantly augmented the expression of osteoblast-specific alkaline phosphatase (10%) and the development of calcium deposits during the advanced stages of osteogenesis (11%) compared to the untreated Ti6Al4V alloy. However, our studies also revealed the positive effect of the pure titanium coating on osteogenic differentiation. These findings should be investigated further and could be employed in the production of endoprosthetic manufacturing. To sum up, the combination of its osteoinductive properties with the ability to enhance cell proliferation and absence of cytotoxic effects positions the tantalum porous coating with MPS as a highly promising candidate for application in bone grafting that demands high biocompatibility and corrosion resistance.

Overall, the research results are encouraging for the medical field, particularly in the context of developing more reliable and effective endoprosthetic implants. The ability to tailor such coatings using MPS technology not only meets the industry standards but also opens avenues for further customization based on specific medical requirements. Targeted recommendations for the surface treatment of titanium alloys in the production of medical endoprosthetic implants suggest a significant advancement in the field of biomaterials and their application in regenerative medicine.

**Author Contributions:** Conceptualization, D.A., A.O., S.V. and Y.S.; methodology, B.A., A.K., A.S., A.N. and H.I.Y.; software, A.K. and B.A.; validation, Y.S., R.Y., S.V. and D.A.; formal analysis, D.A., S.V., A.O., Y.S. and F.K.; investigation, A.N., A.S., A.K., R.Y., H.I.Y., B.A., A.O. and S.V.; resources, A.O.; data curation, D.A.; writing—original draft preparation, D.A.; writing—review and editing, A.O., F.K. and S.W.; visualization, A.O.; supervision, D.A. and S.W.; project administration, Y.S.; funding acquisition, Y.S. All authors have read and agreed to the published version of the manuscript.

**Funding:** This research was funded by Ministry of Science and Higher Education of the Republic of Kazakhstan, grant number AP14869862.

**Institutional Review Board Statement:** Not applicable.

**Informed Consent Statement:** Not applicable.

**Data Availability Statement:** The data presented in this study are available on request from the corresponding author.

**Acknowledgments:** The authors express their gratitude to the current "Ostpartnerschaften" program between Brandenburg University of Technology (BTU) Cottbus-Senftenberg, Germany; De Montfort University (DMU), United Kingdon; Kocaeli University, Turkey; and D. Serikbayev East Kazakhstan State Technical University, Kazakhstan, within which international cooperation and discussion of the research results took place.

**Conflicts of Interest:** The authors declare no conflicts of interest.

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
