# Peer review of "Biocompatibility and Corrosion of Microplasma-Sprayed Titanium and Tantalum Coatings versus Titanium Alloy"

_coatings, doi:10.3390/coatings14020206_

Round 1

Reviewer 1 Report

Comments and Suggestions for Authors

The manuscript coatings-2826780 Round 1

Dr. Darya Alontseva et al. manuscript is devoted to investigates the in-vitro biocompatibility and corrosion resistance of gasabrasive treated Ti6Al4V alloy and microplasma-sprayed unalloyed titanium and tantalum coatings on this alloy. The work has a practical orientation and is saturated with experimental and theoretical methods. However, it has a number of significant shortcomings and cannot be published in this form.

1.         The keywords should not be repeated in the title and introduction.

2.         In the introduction, the authors note that titanium is actively used for implants. The authors should pay attention to the surface treatment of titanium for implants. This is a key point.

3.         From titanium, the authors jump to tantalum and alloy. They do not explain why titanium should be replaced by tantalum or its alloy. What are the advantages of these materials over titanium?

4.         Line 121-123. The authors note that porous roughness is critical for implant engraftment. The authors should mention the work on nanostructured porous oxidization of tantalum 10.3390/ma16030993 and 10.1142/S0218625X21500554. Such treatment will allow to create a protective column-like oxide layer on the tantalum surface and at the same time to create an ordered rough surface.

5.         In the introduction, the authors should justify the choice of the surface treatment method. What are its advantages over others. Why this particular method was chosen.

6.         Despite the large number of co-authors, the article is designed rather carelessly. For example, abbreviations are introduced several times. The authors should be more attentive to the preparation of the manuscript.

7.         Table 1,2,3. Does the reader have to sit and sum the impurity content to get the percentages of tantalum and titanium? Or was this done on purpose so that the purity of the materials used would not be obvious at first glance? In that case, the manuscript should have been rejected immediately.

8.         How does the cleanliness of materials affect the outcome when using MPS? Can dirty materials be used, and what threshold of cleanliness is considered dirty? Authors should note this with references.

9.         Line 251. What is a metallographic microscope? And how does it differ from an optical microscope?

10.      Fig1. It is rather difficult to call it ordered porosity.

11.      In Figure 1, the authors show the cross-sections. And then they count the porosity in Table 6. But porosity can be estimated from a surface with a sufficiently large area. The authors should present a photo of the surface on which they calculated porosity. It is impossible to estimate the porosity objectively from spalls of this size.

12.      Authors should present a methodology for estimating and calculating porosity in Section 2. Since it's questionable.

13.      The authors introduce the object of the statement and abbreviations but do not adhere to it. MPS, microplasma-sprayed, plasma-sprayed, etc. This is misleading. This is a typical mistake of beginning authors.

14.      The figs are of low quality. Fig 2 is smudged, blurred. Line 392-394 It seems like a mockery, because you can't see anything in the Fig.

15.      There's a lot of confusion in the article. It is still not clear on what surface tantalum and titanium were applied. At first I thought it was an alloy. That the alloy was used as a substrate. However, the caption to figure 3 and some other places in the text change my mind about the other "Ti substraite". Presumably there was some other kind of titanium substrate? Why is it not described in the methodological part? Also, there are too many "of" throughout the text, and where they are really needed they are not there. The caption to Figure 3 lacks "of" and has extra ones.

16.      Tafel's graphs in Fig 5 are carelessly done.

17.      Line 465. Of course the metals will oxidize. It is very strange that the authors argue about some corrosion resistance of pure metals. As I noted above, the surface of implants should be protected by oxide, preferably nanostructured. The authors should note this in the discussions and refer to the proposed works.

18.      The design of figs and their layout on the page is terrible. Authors should pay attention to this. Figures are the face of the article. In general, the work is very interesting and can be published if the authors take my comments seriously and take everything into account.

Comments on the Quality of English Language

There are grammatical errors. For example, line 402 is missing "of". In addition, the sentences are not constructed according to English, frequent use of the particle "of". I would recommend the authors to work on the style.

Reviewer 2 Report

Comments and Suggestions for Authors

This research delves into the in-vitro biocompatibility and corrosion resistance of gas abrasive-treated Ti6Al4V alloy, as well as microplasma-sprayed unalloyed titanium and tantalum coatings applied to this alloy. The methodology is thorough, and the results are promising. However, there are a few minor revisions to consider.

How did the selected microplasma spray parameters contribute to the desired porosity and roughness characteristics of the coatings?

what were the key findings regarding in-vitro biocompatibility, corrosion resistance, and adhesive strength between coatings and substrates?

How did the study highlight the critical impact of material selection on biocompatibility and corrosion resistance, with tantalum coatings demonstrating superior cell proliferation, osteogenic differentiation, and corrosion performance compared to titanium coatings and Ti6Al4V alloy?

Reviewer 3 Report

Comments and Suggestions for Authors

The paper is well written and shows significant results. The English is excellent, but the studies were not executed very well. Therefore, the following issues should be addressed prior to publication.

1) The Abstract is vague. Although the experimental plan is well prepared, it is not a new piece of information that tantalum forms oxide layers which exhibit higher corrosion resistance than titanium, e.g. Zhou et al. Materials Science and Engineering A 398 (2005) 28–36. Even the authors themselves sum up the corrosion studies with ”From this point of view, it is feasible to claim that the results produced in this study are compatible with the literature.” and mention the high resistance of tantalum oxide to corrosion, “One of the main reasons for the excellent corrosion properties of Ta is the naturally occurring Ta2O5 (5 µm thick) surface oxide layer”. Hence, writing such well-known facts in the Abstract is not acceptable.

Instead, it would be beneficial to stress the novelty and acheievement of the paper, which is the improved properties of these coating due to the application of microplasma spraying.

2) It is odd that in the cell viability studies in the light of error bars which are three times as high as those for titanium, the experiments were not repeated. I understand that the standard procedure is repeated thrice, but these error bars are not acceptable and the experiment should repeated to at least get a larger number of samples for calculations. Is there an outlier which was incorporated in the calculations or are these values really scattered randomly withn a wider range? This should be checked.

3) The authors state that Ta coating and Ti coating have ”up to 20% pore sizes up to 200 µm” and up to 22% pore sizes up to 200 µm”, respectively. Is this a significant difference? What is the measurement error? The authors write that “similar values of coating porosity were achieved for both coatings”, but do not provide the measurement error.

Comments on the Quality of English Language

Linguistic errors: one sentence is not a paragraph. Please change these, e.g. combine them with the paragraphs below. Lines: 381-385, 386-388 and Line 389 are all single sentences formatted like paragraphs.

Reviewer 4 Report

Comments and Suggestions for Authors

The present manuscript is a study that investigates the in vitro biocompatibility and corosion resistance including other parameters, morphological for instance, of titanium alloys implants coated with pure titanium or tantalum. The entire study is of a very high scientific standard, the text is clear and all procedures and results are described in detail. The results are a significant contribution to the field of research and practical application. It is very difficult, if not impossible, to find the slightest flaw in the study and for this reason I agree to publish it as the manuscript is now presented.

Author Response

   Thank you very much for your thorough review of our manuscript and for your encouraging comments. We are immensely grateful for your recognition of the scientific standard of our work, the clarity of our text, and the detailed description of our procedures and results. It is heartening to hear that our study is seen as a significant contribution to both research and practical application in the field of titanium alloy implants.

            We are honored by your agreement to publish our manuscript as it currently stands. Your support and positive evaluation of our work are greatly motivating and will undoubtedly inspire us in our future research endeavors.

         Thank you once again for your valuable insights and for your recommendation to publish our manuscript. We look forward to the possibility of our research contributing to the advancement of the field.

Round 2

Reviewer 1 Report

Comments and Suggestions for Authors

I recommend the work for publication. The authors were attentive to my comments. 

Comments on the Quality of English Language

I recommend the authors to work on their style.

Reviewer 3 Report

Comments and Suggestions for Authors

The manuscript has been revised in accordance with the recommendation and should be accepted for publication. For the future: problems with large error bars may be countered by performing more measurements (the larger number of points will help find outliers -this is not the case for 3 points, where the test for outliers will rarely give a value that justifies rejection).